# Management of Children and Adolescents with Chest Trauma in Pediatric and Non-Pediatric Departments—A Claims Data Analysis

**DOI:** 10.3390/children10030512

**Published:** 2023-03-05

**Authors:** Peter Zimmermann, Sebastian Kraemer, Nicolas Pardey, Stefan Bassler, Jona T. Stahmeyer, Martin Lacher, Jan Zeidler

**Affiliations:** 1Department of Pediatric Surgery, University of Leipzig, 04103 Leipzig, Germany; 2Section of Thoracic Surgery, Department of Visceral-, Transplant-, Thoracic and Vascular Surgery, University of Leipzig, 04103 Leipzig, Germany; 3Center for Health Economics Research Hannover (CHERH), 30159 Hannover, Germany; 4AOK PLUS-Die Gesundheitskasse fuer Sachsen und Thueringen, 01067 Dresden, Germany; 5AOK-Die Gesundheitskasse fuer Niedersachsen, 30519 Hannover, Germany

**Keywords:** pediatric chest trauma, pediatric thoracic trauma, chest injury, CT imaging, pediatric departments, claims data

## Abstract

Background: To investigate the management of children and adolescents with isolated and combined chest trauma in pediatric (PD) and non-pediatric departments (non-PD). Methods: Anonymized claims data were provided by two large German statutory health insurance funds, covering 6.3 million clients over a 10-year period (2010–2019). Data were extracted for patients who had an inpatient ICD diagnosis of section S20–S29 (injuries to the thorax) and were ≤18 years of age. Demographic and clinical data were analyzed. Results: A total of 4064 children and adolescents with chest trauma were included (mean age 12.0 ± 5.0 years; 55% male). In 1928 cases (47.4%), treatment was provided at PD. Patients admitted to PD underwent CT imaging less frequently (8.1%; non-PD: 23.1%; *p* < 0.0001). Children with a chest drain treated at university/maximum care hospitals (UM) showed more injuries involving multiple body regions compared with non-UM (25.8% vs. 4.5%; *p* = 0.0061) without a difference in the length of hospital stay. Conclusion: Children and adolescents with chest trauma are treated almost equally often in pediatric and adult departments. CT is significantly less frequently used in pediatric departments. Patients with a chest drain treated at a UM showed more concomitant injuries without a longer hospital stay. However, the clinical validity of this finding is questionable.

## 1. Introduction

For children older than 1 year of age, trauma is the leading cause of death [1,2]. The epidemiology of childhood trauma varies based on factors such as the child’s age, injury mechanism, and region. The most common causes of childhood injuries include falls, road traffic accidents, and sports-related injuries [3,4]. In the United States, assault is also a relevant cause of pediatric trauma [5]. Chest trauma accounts for 10% of all trauma-related injuries in children and 14% of pediatric trauma-related deaths [6,7,8,9]. Because of anatomical and physiological differences, children with thoracic injuries present different injury patterns than adults [8,10,11]. Evaluating pediatric trauma patients can be difficult because of physiological and developmental factors that vary with age, changes in mental condition, and the discomfort that inexperienced medical providers may experience when caring for children [12,13]. Research indicates that injured children who receive medical care at adult trauma centers have a notably higher in-hospital mortality compared with those treated at pediatric trauma centers [14,15]. Therefore, in this observational cohort study, we analyzed the management of children and adolescents with chest trauma in pediatric and non-pediatric departments.

## 2. Materials and Methods

In this observational cohort study, anonymized claims data from the years 2010–2019 were obtained from two large public health insurance providers in Germany, covering 6.3 million clients (1 million children) [16].

The diagnosis and procedures were categorized using the German adaptations of the International Classification of Diseases in its 10th revision (ICD-10-GM) and the International Classification of Procedures in Medicine (ICPM), which is also known as the Operation and Procedure Classification System (OPS).

Data analysis was carried out in compliance with the Good Practice Secondary Data Analysis (GPS) [17]. All patients with inpatient International Classification of Diseases (ICD) codes section S20–S29 (injuries to the thorax) aged ≤18 years who had been continuously insured members of their statutory health insurance for at least one year after their index diagnosis or until death were included. Patient characteristics including demographics (age and gender), as well as the characteristics of inpatient treatment such as type of department (pediatrics, pediatric surgery, general surgery, and orthopedic and trauma surgery), type of hospital (university/maximum care hospital (UM) vs. non-UM), use of thoracic CT and/or thoracic MRI (OPS codes 3-202, 3-222/3-809, 3-822), length of hospital stay, and specific comorbidities were evaluated. The presence of a specific traumatic- or injury-related comorbidity was assumed if ICD-10-GM codes S00-S99, T00-T07, T08-T14, T66-T78, T79, R57.1, excl. S20–S29 were documented. According to the ICPM, a subgroup analysis of patients with chest drain insertion (OPS codes 5-340.0 or 8-144) was performed. Data analysis included descriptive statistics and multivariate regression models. For categorical variables, differences between cohorts regarding the characteristics of the study population and outcomes were analyzed using Chi-Squared test or Fisher exact test and Mann–Whitney-U test and Kruskal–Wallis tests for continuous variables. To check for homogeneity of variance and multicollinearity, the Levene test and variance inflation factor were used. Finally, demographic and clinical data were analyzed in a logistic regression to determine the factors influencing the use of thoracic imaging and chest drain insertion. Data analyses were performed using SAS 9.4 for Windows, SAS Institute Inc., Cary, NC, USA.

## 3. Results

A total of 4064 children and adolescents with chest trauma (mean age 12.0 ± 5.0 years; 55.4% male) were included (Table 1 and Table 2). Treatment was provided by different pediatric (PD) and non-pediatric departments (non-PD) (PD: pediatrics: N = 1519, 39.1%; pediatric Surgery: N = 409, 10.5%; non-PD: general surgery: N = 1223, 31.5%; orthopedic and trauma surgery: N = 731, 18.8%; general medicine/unspecified surgical: N = 182, 4.5%) (Table 1 and Table 3).

In 1928 cases (47.4%), treatment was provided at PD (Table 1). Children up to 11 years of age were significantly more often treated in PD (N = 1146, 59.4%) than in non-PD (N = 453, 21.2%) (*p* < 0.0001) (Table 1). In the age group of 12–18 years, more patients were admitted to non-PD than PD (N = 1683, 78.8% vs. N = 782 (40.6%); *p* < 0.0001) (Table 1). In total, 70% (N = 1683) of all patients aged 12–18 years (N = 2465) were treated in adult trauma centers.

Extra-thoracic injuries were found in 2790 (68.7%) patients, with a significant higher rate in non-PD (N = 1550, 72.6%) compared with PD (N = 1240, 64.3%) (*p* = 0.0018) (Table 1). Injuries to the extremities were the most common extra-thoracic injuries (N = 1386, 34.1%), followed by injuries to the abdomen (N = 1324, 32.6%) and injuries to the head (N = 1254, 30.9%).

There was no difference regarding the rate of injuries involving multiple body regions between pediatric trauma patients treated in PD vs. non-PD (total: N = 130, 3.2%; PD: N = 63, 3.3%; non-PD: N = 67, 3.1%; P = 0.8085) (Table 1).

In 479 (11.8%) cases, care was provided at a UM (Table 4). Concerning the type of hospital, there was a significantly higher frequency of extra-thoracic injuries in the patients admitted to UM (75.8% vs. 67.7% (non-UM); P = 0.0474) (Table 3). Injuries involving multiple body regions were significantly more common in UM patients (total: N = 130, 3.2%; UM: N = 44, 9.2%; non-UM: N = 86, 2.4%; *p* < 0.0001) (Table 4).

For children and adolescents with a chest drain (CD), the length of hospital stay between UM and non-UM did not show a significant difference in univariate analysis (14.9 ± 14.3 vs. 20.8 ± 26.8 days; *p* = 0.6518) (Table 5), as well as in multivariate regression (OR: 0.983 [0.955–1.005]; *p* = 0.1786) (Table 6).

Children admitted to PD underwent thoracic CT imaging significantly less frequently (8.1%) than patients in non-PD (23.1%; *p* < 0.0001) (Table 1). Multivariate logistic regression confirmed a significantly higher odds ratio (OR) for the use of CT by non-PD (OR: 2.429 [95-%-CI: 1.937–3.062]; *p* < 0.0001) (Table 7). Other factors associated with the use of CT included age between 12 and 18 years, male gender, admission to a UM (vs. non-UM), length of hospital stay, and traumatic non-thoracic comorbidity (Table 7).

In total, 40 patients died (mean age 11.2 ± 6.3 years; 60.0% male) (Table 8). Care was provided at PD in 12 cases (30.0%); 18 (45.0%) children were treated at UM (Table 8). In the subgroup of deceased children who were managed in adult trauma centers (N = 28, 70.0%) six were younger than 12 years (<1 year: N = 1, 3.6%; 1 to 5 years: N = 3, 10.7%; 6 to 11 years: N = 2, 7.1%) (Table 8). The mean length of hospital stay until death was 5.0 ± 12.9 days (PD: 3.4 ± 3.9 d; non-PD: 5.7 ± 15.2 d). Because of the low number of patients and lack of significance, no further analysis was carried out.

**Table 1 children-10-00512-t001:** Patient characteristics (pediatric departments/non-pediatric departments).

	All Departments	Pediatric Departments	Non-Pediatric Departments	*p* †
**Total patients [n (%)]**	4064	1928 (47.4)	2136 (52.6)	
**Sex**				0.0405
Female [n (%)]	1814 (44.6)	893 (46.3)	921 (43.1)	
Male [n (%)]	2250 (55.4)	1035 (53.7)	1215 (56.9)	
**Mean age (years)**	12.0 ± 5.0	9.6 ± 5.1	14.2 ± 3.8	<0.0001
**Age [years]**				
<1 [n (%)]	80 (2.0)	78 (4.1)	2 (0.1)	<0.0001
1 to 5 [n (%)]	491 (12.1)	401 (20.8)	90 (4.2)	<0.0001
6 to 11 [n (%)]	1028 (25.3)	667 (34.6)	361 (16.9)	<0.0001
12 to 18 [n (%)]	2465 (60.7)	782 (40.6)	1683 (78.8)	<0.0001
**Type of hospital**				0.0032
UM [n (%)]	479 (11.8)	197 (10.2)	282 (13.2)	
Non-UM [n (%)]	3585 (88.2)	1731 (89.8)	1854 (86.8)	
**Imaging**				
CT [n (%)]	650 (16.0)	156 (8.1)	494 (23.1)	<0.0001
MRI [n (%)]	21 (0.5)	14 (0.7)	7 (0.3)	0.0840
CT or MRI [n (%)]	671 (16.5)	170 (8.8)	501 (23.5)	<0.0001
Without	3393 (83.5)	1758 (91.2)	1635 (76.5)	<0.0001
**Mean length of hospital stay [days]**	5.0 ± 12.2	4.4 ± 9.5	5.5 ± 14.2	0.2887
**Comorbidity ***				
Any	2790 (68.7)	1240 (64.3)	1550 (72.6)	0.0018
S00–S09	1254 (30.9)	602 (31.2)	652 (30.5)	0.6672
S10–S19	536 (13.2)	178 (9.2)	358 (16.8)	<0.0001
S30–S39	1324 (32.6)	538 (27.9)	786 (36.8)	<0.0001
S40–S99	1386 (34.1)	560 (29.0)	826 (38.7)	<0.0001
T00–T07	130 (3.2)	63 (3.3)	67 (3.1)	0.8085
T08–T14	94 (2.3)	61 (3.2)	33 (1.5)	0.0007
T66–T78	65 (1.6)	59 (3.1)	6 (0.2)	<0.0001
T79	61 (1.5)	16 (0.8)	45 (2.1)	0.0009
R57.1	17 (0.4)	3 (0.2)	14 (0.7)	0.0147

Characteristics of 4064 patients with ICD-10-GM code S20–S29 (injuries to the thorax) aged ≤ 18 years. UM = university/maximum care hospital (UM). Non-UM = non-university/maximum care hospital. CT = computed tomography. MRI = magnetic resonance imaging. * Extra-thoracic injuries; ICD-10-GM = International Classification of Diseases in its 10th version, German Modification (Table 7): Comorbidities/Extra-thoracic injuries (ICD-10-GM)). † pediatric departments vs. non-pediatric departments.

**Table 2 children-10-00512-t002:** Comorbidities/extra-thoracic injuries (ICD-10-GM).

S00–S09	Injuries to the head
S10–S19	Injuries to the neck
S30–S39	Injuries to the abdomen
S40–S99	Injuries to the extremities
T00–T07	Injuries involving multiple body regions
T08–T14	Injuries to unspecified parts of the trunk, extremities, or other body regions
T66–T78	Other and unspecified effects of external causes
T79	Certain early complications of trauma, not elsewhere classified
R57.1	Hypovolemic shock
S00–S09	Injuries to the head

ICD-10-GM = International Classification of Diseases in its 10th version, German Modification (https://www.bfarm.de/EN/Code-systems/Classifications/ICD/ICD-10-GM/Code-search/_node.html, accessed on 27 February 2023).

**Table 3 children-10-00512-t003:** Patient characteristics in different departments.

	All Departments	Pediatrics	Pediatric Surgery	General Surgery	Orthopedic & Trauma Surgery	*p* †
**Total patients [n (%)]**	3882 ^a^	1519 (39.1)	409 (10.5)	1223 (31.5)	731 (18.8)	
**Sex**						0.0572
Female [n (%)]	1735 (44.7)	719 (47.3)	174 (42.5)	520 (42.5)	322 (44.0)	
Male [n (%)]	2147 (55.3)	800 (52.7)	235 (57.5)	703 (57.5)	409 (56.0)	
**Mean age (years)**	12.0 ± 5.0	9.7 ± 5.2	9.3 ± 4.6	14.3 ± 3.6	14.1 ± 4.0	<0.0001
**Age [years]**						
1 [n (%)]	79 (2.0)	62 (4.1)	16 (3.9)	1 (0.1)	-	<0.0001
1 to 5 [n (%)]	479 (12.3)	322 (21.2)	79 (19.3)	40 (3.3)	38 (5.2)	<0.0001
6 to 11 [n (%)]	1007 (25.9)	502 (33.1)	165 (40.3)	213 (17.4)	127 (17.4)	<0.0001
12 to 18 [n (%)]	2317 (59.7)	633 (41.7)	149 (36.4)	969 (79.2)	566 (77.4)	<0.0001
**Type of hospital**						<0.0001
UM [n (%)]	444 (11.4)	80 (5.3)	117 (28.6)	65 (5.3)	182 (24.9)	
Non-UM [n (%)]	3438 (88.6)	1439 (94.7)	292 (71.4)	1158 (94.7)	549 (75.1)	
**Imaging**						
CT [n (%)]	574 (14.8)	97 (6.4)	59 (14.4)	193 (15.8)	225 (30.8)	<0.0001
MRI [n (%)]	21 (0.5)	7 (0.5)	7 (1.7)	3 (0.3)	4 (0.5)	0.0151
CT or MRI [n (%)]	595 (15.3)	104 (6.8)	66 (16.1)	196 (16.0)	229 (31.3)	<0.0001
Without	3287 (84.7)	1415 (93.2)	343 (83.9)	1027 (84.0)	502 (68.7)	<0.0001
**Mean length of hospital stay [days]**	4.8 ± 11.0	3.5 ± 7.9	7.7 ± 13.3	3.5 ± 5.5	4.9 ± 8.6	<0.0001
**Comorbidity ***						
Any	2659 (68.5)	946 (62.3)	294 (71.9)	868 (71.0)	551 (75.4)	0.0014
S00–S09	1157 (29.8)	452 (29.8)	150 (36.7)	323 (26.4)	232 (31.7)	0.0062
S10–S19	518 (13.3)	138 (9.1)	40 (9.8)	213 (17.4)	127 (17.4)	<0.0001
S30–S39	1273 (32.8)	379 (25.0)	159 (38.9)	442 (36.1)	293 (40.1)	<0.0001
S40–S99	1323 (34.1)	431 (28.4)	129 (31.5)	467 (38.2)	296 (40.5)	<0.0001
T00–T07	115 (3.0)	37 (2.4)	26 (6.4)	35 (2.9)	17 (2.3)	0.0003
T08–T14	89 (2.3)	38 (2.5)	23 (5.6)	16 (1.3)	12 (1.6)	<0.0001
T66–T78	62 (1.6)	42 (2.8)	17 (4.2)	3 (0.2)	-	<0.0001
T79	51 (1.3)	10 (0.7)	6 (0.2)	14 (1.1)	21 (2.9)	0.0006
R57.1	7 (0.2)	3 (0.2)	-	2 (0.2)	2 (0.3)	-

Characteristics of 3882 patients with ICD-10-GM code S20–S29 (injuries to the thorax) aged ≤ 18 years. UM = university/maximum care hospital (UM). Non-UM = non-university/maximum care hospital. CT = computed tomography, MRI = magnetic resonance imaging. * Extra-thoracic injuries; ICD-10-GM = International Classification of Diseases in its 10th version, German Modification (Table 7): Comorbidities/Extra-thoracic injuries (ICD-10-GM)). † Comparison between the different departments. ^a^ Due to exceptionally low patient numbers, the data of children admitted to internal medicine departments or other unspecified surgical departments were not further analyzed due to reasons related to data protection and insufficient statistical power (n = 182).

**Table 4 children-10-00512-t004:** Patient characteristics (UM/non-UM).

	All Departments	UM	Non-UM	*p* †
**Total patients [n (%)]**	4064	479 (11.8)	3585 (88.2)	
**Sex**				0.0526
Female [n (%)]	1814 (44.6)	194 (40.5)	1620 (45.2)	
Male [n (%)]	2250 (55.4)	285 (59.5)	1965 (54.8)	
**Mean age (years)**	12.0 ± 5.0	11.4 ± 5.4	12.1 ± 5.0	0.0093
**Age [years]**				
<1 [n (%)]	80 (2.0)	18 (3.8)	62 (1.7)	0.0030
1 to 5 [n (%)]	491 (12.1)	70 (14.6)	421 (11.7)	0.0915
6 to 11 [n (%)]	1028 (25.3)	121 (25.3)	907 (25.3)	0.9766
12 to 18 [n (%)]	2465 (60.7)	270 (56.4)	2195 (61.2)	0.1926
**Department**				0.0032
Pediatric department [n (%)]	1928 (47.4)	197 (41.1)	1731 (48.3)	
Non-Pediatric department [n (%)]	2136 (52.6)	282 (58.9)	1854 (51.7)	
**Imaging**				
CT [n (%)]	650 (16.0)	146 (30.5)	504 (14.1)	<0.0001
MRI [n (%)]	21 (0.5)	6 (1.3)	15 (0.4)	0.0305
CT or MRI [n (%)]	671 (16.5)	152 (31.7)	519 (14.5)	<0.0001
Without	3393 (83.5)	327 (68.3)	3066 (85.5)	<0.0001
**Mean length of hospital stay [days]**	5.0 ± 12.2	8.5 ± 13.3	4.5 ± 12.0	<0.0001
**Comorbidity ***				
Any	2790 (68.7)	363 (75.8)	2427 (67.7)	0.0474
S00–S09	1254 (30.9)	198 (41.3)	1056 (29.5)	<0.0001
S10–S19	536 (13.2)	57 (11.9)	479 (13.4)	0.4029
S30–S39	1324 (32.6)	180 (37.6)	1144 (31.9)	0.0429
S40–S99	1386 (34.1)	187 (39.0)	1199 (33.4)	0.0509
T00–T07	130 (3.2)	44 (9.2)	86 (2.4)	<0.0001
T08–T14	94 (2.3)	17 (3.5)	77 (2.1)	0.0589
T66–T78	65 (1.6)	18 (3.8)	47 (1.3)	<0.0001
T79	61 (1.5)	22 (4.6)	39 (1.1)	<0.0001
R57.1	17 (0.4)	2 (0.4)	15 (0.4)	1

Characteristics of 4064 patients with ICD-10-GM code S20–S29 (injuries to the thorax) aged ≤ 18 years. UM = university/maximum care hospital (UM) Non-UM = non-university/maximum care hospital. CT = computed tomography. MRI = magnetic resonance imaging. * Extra-thoracic injuries; ICD-10-GM = International Classification of Diseases in its 10th version, German Modification (Table 7): Comorbidities/Extra-thoracic injuries (ICD-10-GM)). † university hospital vs. non-university hospital.

**Table 5 children-10-00512-t005:** Patient characteristics (subgroup with OPS 5-340.0 or 8-144, drainage of the pleural cavity).

	All	UM	Non-UM	*p* †
**Total patients [n (%)]**	98	31 (31.6)	67 (68.4)	
**Sex**				0.2972
Female [n (%)]	25 (25.5)	10 (32.3)	15 (22.4)	
Male [n (%)]	73 (74.5)	21 (67.7)	52 (77.6)	
**Mean age (years)**	13.2 ± 5.5	12.7 ± 5.3	13.4 ± 5.7	0.2595
**Age [years]**				
0 to 5 [n (%)]	14 (14.3)	4 (12.9)	10 (14.9)	0.5307
6 to 11 [n (%)]	12 (12.2)	6 (19.4)	6 (9.0)	0.3278
12 to 18 [n (%)]	72 (73.5)	21 (67.7)	51 (76.1)	0.6570
**Department**				0.3028
Pediatric department [n (%)]	28 (28.6)	11 (35.5)	17 (25.4)	
Non-Pediatric department [n (%)]	70 (71.4)	20 (64.5)	50 (74.6)	
**Imaging**				
CT [n (%)]	65 (66.3)	23 (74.2)	42 (62.7)	0.5116
MRI [n (%)]	3 (0.5)	1 (3.2)	2 (3.0)	-
CT or MRI [n (%)]	68 (69.4)	24 (77.4)	44 (65.7)	0.5123
Without	30 (30.6)	7 (22.6)	23 (34.3)	0.3301
**Mean length of hospital stay [days]**	19.0 ± 23.6	14.9 ± 14.3	20.8 ± 26.8	0.6518
**Comorbidity ***				
Any	68 (69.4)	26 (83.9)	42 (62.7)	0.2392
S00–S09	42 (42.9)	13 (41.9)	29 (43.3)	0.9281
S10–S19	9 (9.2)	4 (12.9)	5 (7.5)	0.4768
S30–S39	44 (44.9)	15 (48.4)	29 (43.3)	0.7223
S40–S99	37 (37.8)	12 (38.7)	25 (37.3)	0.9133
T00–T07	11 (11.2)	8 (25.8)	3 (4.5)	0.0061
T08–T14	3 (3.1)	1 (3.2)	2 (3.0)	-
T66–T78	2 (2.0)	1 (3.2)	1 (1.5)	-
T79	18 (18.4)	8 (25.8)	10 (14.9)	0.1785
R57.1	5 (5.1)	1 (3.2)	4 (6.0)	-

Characteristics of 98 patients with ICD-10-GM code S20–S29 (injuries to the thorax) and OPS codes 5-340.0 or OPS 8-144 (drainage of the chest wall or pleural cavity; therapeutic drainage of pleural cavity) aged ≤18 years. OPS-Code 5-340.0 Operations on chest wall, pleura, mediastinum, and diaphragm: Incision into chest wall and pleura: Drainage of the chest wall or pleural cavity, open surgical (https://gesund.bund.de/en/ops-code-search/5-340, accessed on 27 February 2023); OPS-Code 8-144 Other forms of therapeutic catheterization and cannulation: Therapeutic drainage of pleural cavity (https://gesund.bund.de/en/ops-code-search/8-144, accessed on 27 February 2023). UM = university/maximum care hospital (UM). Non-UM = non-university/maximum care hospital. CT = computed tomography. MRI = magnetic resonance imaging. * Extra-thoracic injuries; ICD-10-GM = International Classification of Diseases in its 10th version, German Modification (Table 7): Comorbidities/Extra-thoracic injuries (ICD-10-GM)). † university hospital vs. non-university hospital.

**Table 6 children-10-00512-t006:** Estimated odds ratio for patients treated at UM vs. non-UM (subgroup with OPS 5-340.0 or 8-144, drainage of the pleural cavity).

	OR	95%-CI	*p*
**Age Group**			
**6 to 11**	1.234	0.213–7.454	0.8142
**12 to 18**	1.200	0.236–6.717	0.8273
**Male gender**	0.609	0.214–1.761	0.3516
**Length of Hospital stay**	0.983	0.955–1.005	0.1768
**Any Comorbidity ***	3.544	1.198–12.427	0.0312
**Pediatric Department**	0.434	0.108–1.742	0.2310

Estimated odds ratio for patients with injuries to the thorax and drainage of the pleural cavity at UM vs. non-UM. UM = university/maximum care hospital (UM). Non-UM = non-university/maximum care hospital. OPS-Code 5-340.0 Operations on chest wall, pleura, mediastinum and diaphragm: Incision into chest wall and pleura: Drainage of the chest wall or pleural cavity, open surgical (https://gesund.bund.de/en/ops-code-search/5-340, accessed on 27 February 2023); OPS-Code 8-144 Other forms of therapeutic catheterization and cannulation: Therapeutic drainage of pleural cavity (https://gesund.bund.de/en/ops-code-search/8-144, accessed on 27 February 2023) * aggregation of ICD-10-GM S00-S99, T00-T07, T08-T14, T66-T78, T79, R57.1, excl. S20-S29. CI = confidence interval. ICD-10-GM = International Classification of Diseases in its 10th version, German Modification. OR = odds ratios. Reference categories: 0 to 5, female gender, no comorbidity, non-pediatric department.

**Table 7 children-10-00512-t007:** Estimated odds ratio for use of CT vs. no CT.

	OR	95%-CI	*p*
**Age Group**			
**<1**	0.042	0.005–0.209	0.0007
**6 to 11**	0.967	0.632–1.501	0.8776
**12 to 18**	1.866	1.275–2.799	0.0018
**Male gender**	1.624	1.329–1.990	<0.0001
**University Hospital**	2.289	1.762–2.961	<0.0001
**Length of Hospital stay**	1.076	1.062–1.091	<0.0001
**Any Comorbidity ***	3.409	2.592–4.553	<0.0001
**Non-Pediatric Department**	2.429	1.937–3.062	<0.0001

* aggregation of ICD-10-GM S00–S99, T00–T07, T08–T14, T66–T78, T79, R57.1, excl. S20–S29. CI = confidence interval, ICD-10-GM = International Classification of Diseases in its 10th version, German Modification. OR = odds ratios. Reference categories: 1 to 5, female gender, non-university hospital, no comorbidity, pediatric department.

**Table 8 children-10-00512-t008:** Patient characteristics (subgroup of deceased patients).

	All Departments	Pediatric Departments	Non-Pediatric Departments
**Total patients [n (%)]**	40	12 (30.0)	28 (70.0)
**Sex**			
Female [n (%)]	16 (40.0)	6 (50.0)	10 (35.7)
Male [n (%)]	24 (60.0)	6 (50.0)	18 (64.3)
**Mean age (years)**	11.2 ± 6.3	5.5 ± 4.8	13.6 ± 5.2
**Age [years]**			
<1 [n (%)]	3 (7.5)	2 (16.7)	1 (3.6)
1 to 5 [n (%)]	8 (20.0)	5 (41.7)	3 (10.7)
6 to 11 [n (%)]	5 (12.5)	3 (25.0)	2 (7.1)
12 to 18 [n (%)]	24 (60.0)	2 (16.7)	22 (78.6)
**Type of hospital**			
UM [n (%)]	18 (45.0)	5 (41.7)	13 (46.4)
Non-UM [n (%)]	22 (55.0)	7 (58.3)	15 (53.6)
**Imaging**			
CT [n (%)]	30 (75.0)	9 (75.0)	21 (75.0)
MRI [n (%)]	-	-	-
CT or MRI [n (%)]	-	-	-
Without	10 (25.0)	3 (25.0)	7 (25.0)
**Mean length of hospital stay [days]**	5.0 ± 12.9	3.4 ± 3.9	5.7 ± 15.2
**Comorbidity ***			
Any	38 (95.0)	11 (91.7)	27 (96.4)
S00–S09	29 (72.5)	10 (83.3)	19 (67.9)
S10–S19	1 (2.5)	-	1 (3.6)
S30–S39	14 (35.0)	3 (25.0)	11 (39.3)
S40–S99	16 (40.0)	5 (41.7)	11 (39.3)
T00–T07	8 (20.0)	4 (33.3)	4 (14.3)
T08–T14	-	-	-
T66–T78	-	-	-
T79	11 (27.5)	1 (8.3)	10 (35.7)
R57.1	6 (15.0)	-	6 (21.4)

Characteristics of 40 deceased patients with ICD-10-GM code S20–S29 (injuries to the thorax) aged ≤ 18 years. Because of the low number of patients and lack of significance, no further analysis was carried out. UM = university/maximum care hospital (UM). Non-UM = non-university/maximum care hospital. CT = computed tomography. MRI = magnetic resonance imaging. * Extra-thoracic injuries; ICD-10-GM = International Classification of Diseases in its 10th version, German Modification (Table 7): Comorbidities/Extra-thoracic injuries (ICD-10-GM)).

## 4. Discussion

Seriously injured children are treated in accordance with the guidelines of the German Society for Pediatric Surgery and the German Society for Trauma Surgery [18]. In this observational cohort study, we set out to evaluate the management of children and adolescents with chest trauma and the extent to which treatment patterns differ at first stay between pediatric and non-pediatric departments [19,20]. Claims data from two large German health insurance funds were used, covering ∼6.3 million clients representing 1 million children. Consequently, one strength of this study is the large study population we were able to analyze. Many studies have demonstrated the ability of claims data to adequately represent pediatric care conditions [12,21,22,23,24,25].


*Adult trauma centers treat most children between 12-18 years*


In this study, children and adolescents with chest trauma were treated almost equally often in pediatric and adult departments (Table 1). However, differentiation according to age groups showed that children up to 11 years of age were significantly more often treated in pediatric departments and that most patients in the age group of 12–18 years were admitted to non-pediatric departments (Table 1).

Research indicates that injured children who received medical care at adult trauma centers had a notably higher in-hospital mortality compared with those treated at pediatric trauma centers [14,26,27]. As most children between 12–18 years were managed in adult trauma centers, our results may emphasize the importance of the centralization of specialized pediatric trauma care not only for younger pediatric trauma patients, but also for injured adolescents.


*Extra-thoracic injuries are found in more than two-third of children with chest trauma*


Extra-thoracic injuries were found in approximately 70% of patients. As chest trauma is attributed to high-energy forces such as motor vehicle accidents and falls from heights, the occurrence of associated injuries was not unexpected and was in line with other studies [8,28]. Generally, head trauma is the most prevalent extra-thoracic injury in patients with chest trauma [29,30,31]. However, in this study injuries of the extremities (34%) were the most common extra-thoracic injuries, followed by abdominal trauma (33%) and head trauma (31%).


*Pediatric surgeons treat most children with chest trauma and multiple extra-thoracic injuries*


There was no significant difference in the frequency of injuries involving multiple body regions between patients treated in pediatric and non-pediatric departments (Table 1). However, further differentiation showed that pediatric surgeons treated most chest trauma patients with injuries involving multiple body regions. Considering the hospital structure in Germany, it can be assumed that children and adolescents with chest trauma admitted to Pediatrics were treated by pediatricians and general surgeons or trauma surgeons (no dedicated pediatric surgery department). The management of these patients by adult surgeons in cooperation with pediatricians may be superior to treatment by adult surgeons alone. However, as there is evidence that trauma management by specialized pediatric trauma surgeons shows better outcomes, children with chest trauma should be admitted to specialized pediatric trauma centers with a pediatric surgery department [14,26,27].


*Only one-eighth of children with chest trauma are treated at a university/maximum care hospital*


Only 12% of children and adolescents with chest trauma in Germany were treated at a university/maximum care hospital (Table 1 and Table 4). These patients showed a significantly higher rate of extra-thoracic injuries and injuries involving multiple body regions compared with the children treated at non-university hospitals (Table 4). This finding may represent a certain triage of patients by the emergency services in the preclinical setting and centralization of severe cases. However, the low percentage of children with thoracic trauma treated at a university/maximum care hospital (12%) may indicate some structural deficits in the specialized trauma care of children with chest trauma.


*Children with a chest drain who are managed at a university/maximum care hospital present more injuries involving multiple body regions*


Less than 3% of the studied patients received a chest drain (Table 5). Children with a chest drain who were treated at a university hospital presented significantly more injuries involving multiple body regions than patients with a chest drain managed at non-university hospitals. However, a significant difference in the length of hospital stay was not observed. The mean length of stay in UM was even lower than in non-UM, but not significantly (Table 5). This may indicate the effectiveness and quality of specialized pediatric trauma care in maximum care hospitals, but, as this subgroup accounted for less than 3% of the treated patients, the clinical validity of this finding is questionable (Table 5).


*The in-hospital mortality of children with chest trauma is extremely low*


At 1% (40 patients), the in-hospital mortality was extremely low (Table 8). However, differentiation according to age groups showed that six of the 28 children who died in adult departments were younger than 12 years, with four of them being younger than six years. As there was no information about the severity of the trauma (e.g., abbreviated injury score) and the initial or subsequent clinical state of injury, a differentiated interpretation of selected cases was strongly limited. Nevertheless, the question remains whether treatment of the six children below 12 years who died in adult departments would have been more successful in designated pediatric trauma centers.


*Adult trauma centers use CT imaging more frequent than pediatric trauma centers*


In adult trauma centers, CT imaging was used more frequently than in pediatric trauma centers (Table 1). This represents a further advantage of specialized pediatric trauma care. Medical professionals who provide care to children daily might be more conscious of the potential long-term effects of radiation exposure, including the increased risk of cancer-related deaths [12,13]. In addition, pediatric departments typically cooperate with pediatric radiologists who ensure that radiation exposure is kept to a minimum [12]. For patients who were treated in a non-pediatric department, the chance for thoracic CT imaging was 2.4-fold higher compared with pediatric departments (Table 7). Additional extra-thoracic injuries increased the likelihood to receive a thoracic CT by 3.4-fold. This might be because patients with a larger comorbidity burden were classified as more severely injured, resulting in a higher rate of thoracic CTs. To examine the comparability of CT imaging between PD and non-PD, logistic regression for the use of CT was performed (Table 9). We estimated the same model as in Table 7, except that an interaction term of the comorbidity and treatment department was added. The non-significance of the interaction term (*p* = 0.2801) suggests that we can assume similar comorbidity patterns in PD and non-PD. Furthermore, the broad distribution of thoracic CT use rate between different departments (Table 3) illustrates the potential for reducing the risk of radiation-induced cancer mortality by implementing evidence-based pediatric trauma guidelines.


*MRI has low relevance in the diagnostic management of pediatric chest trauma*


MRI allows for excellent evaluation and diagnosis of thoracic organ lesions. However, as the MRI use rate was exceptionally low for all patients (0.5%), no further analysis was performed (Table 1 and Table 3).


*Limitations*


We are aware of several limitations of our study. The precision of diagnostic and procedural coding could have been impacted by the coding practices employed by the hospital. Therefore, there is a risk that the results might be influenced by misclassification of the clinical and imaging outcomes. As claims data are collected for billing and reimbursement purposes, important information on clinical parameters, such as the severity of the trauma (e.g., abbreviated injury score), is missing. According to the study design, the treatment history prior to the first stay was unobservable. Because of the absence of information regarding the initial or subsequent clinical condition of injury, it was not possible to entirely reconstruct the clinical decisions made regarding the usage of CT imaging or the insertion of a thoracic drain. Finally, local hospital standards, technical and personnel resources at the time of admission, and the setting in which the primary evaluation of the patient took place may have been important confounders influencing the actions and decisions of the attending physicians.

## 5. Conclusions

Our study describes the current management situation of children and adolescents with chest trauma in pediatric and non-pediatric departments in Germany based on an analysis of claims data. This analysis alone does not have a direct impact on the current management situation. The main limitations are the missing information for clinical variables, e.g., the severity of the trauma (e.g., abbreviated injury score). However, because of the large sample size of 1 million children, our results are very reliable. Thoracic CT imaging is significantly less frequently used in pediatric departments, indicating that the implementation of guidelines based on scientific evidence may lead to a decrease in the utilization of CT, especially in non-pediatric departments. Patients with a chest drain who were treated at a university/maximum care hospital show more injuries involving multiple body regions than patients with a chest drain managed at non-university hospitals, without any difference in the duration of hospitalization. This may indicate the effectiveness and quality of specialized pediatric trauma care in maximum care hospitals. However, as this subgroup accounts for less than 3% of patients, the clinical validity of this finding is questionable. As 50% of children and adolescents with chest trauma are treated in adult departments, our results emphasize the importance of the centralization and implementation of evidence-based guidelines for specialized pediatric trauma care, especially in adult trauma centers. In summary, our study indicates high-quality care for children and adolescents with chest trauma in Germany, as evidenced by the low mortality rate. For further optimization, we suggest implementing evidence-based guidelines, such as those that aim to reduce the risk of ionizing radiation when using CT Thorax. However, linking claims data with clinical data is necessary to improve evaluation. Thus, our findings highlight the need for this connection in future studies.

## Figures and Tables

**Table 9 children-10-00512-t009:** Comparability of PD and non-PD with regard to the performance of CT.

	OR	95%-CI	*P*
**Age Group**			
**<1**	0.046	0.005–0.221	0.0008
**6 to 11**	0.970	0.634–1.508	0.8908
**12 to 18**	1.881	1.284–2.824	0.0016
**Male gender**	1.623	1.328–1.989	<0.0001
**University Hospital**	2.284	1.762–2.961	<0.0001
**Length of Hospital stay**	1.076	1.062–1.091	<0.0001
**Any Comorbidity ***	4.346	2.619–7.687	<0.0001
**Non-Pediatric Department**	3.273	1.848–6.083	<0.0001
**Any Comorbidity *|Non-Pediatric Department ****	0.707	0.367–1.304	0.2801

* aggregation of ICD-10-GM S00–S99, T00–T07, T08–T14, T66–T78, T79, R57.1, excl. S20–S29. ** interaction term of Any Comorbidity and Non-Pediatric Department. CI = confidence interval. ICD-10-GM = International Classification of Diseases in its 10th version, German Modification. OR = odds ratios. Reference categories: 1 to 5, female gender, non-university hospital, no comorbidity, pediatric department.

## Data Availability

Restrictions apply to the availability of these data. Data were obtained from two large German statutory health insurances and data sharing is not applicable.

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
