# Peer review of "Management of Children and Adolescents with Chest Trauma in Pediatric and Non-Pediatric Departments—A Claims Data Analysis"

_children, 2023, doi:10.3390/children10030512_

Round 1

Reviewer 1 Report

The paper presented for the review is aimed to investigate the management of children and adolescents with isolated and combined chest trauma in pediatric and non-pediatric departments.
The authors conducted meticulous analysis of the statistical data from the large database of the health insurance (6,3 million clients). The final analysis included impressive number of patients - more than 4000 children and adolescents with chest trauma. The accumulated data gives the clear prospective of the different dimensions of the pediatric chest trauma including not only frequency of CT but also complications, mortality and many others.
The paper is well structured and clearly presented.
The conclusions are supported by the data, limitations and advantages of the paper are also presented. All the important data are presented in the tables and easy to understand.
The paper makes good impression, contains important information and recommended for the publication in the present version.
The only possible recommendation is to modify the title of the paper according to the aim (investigate the management of children and adolescents with isolated and combined chest trauma in pediatric and non-pediatric departments) and not limit it to the CT issues.

Author Response

Point-by-point response to the comments of the reviewers

The authors would like to thank the reviewers for carefully reading our manuscript and providing valuable comments and suggestions to improve the paper. The following responses have been prepared to address all of the reviewers' comments in a point-by-point style.

Response to Reviewer #1:

We thank reviewer #1 for her/his comments and are delighted that she/he finds our manuscript well-structured and clear. To acknowledge this valuable reviewer comment, we have changed the title of the paper to: “Management of children and adolescents with chest trauma in pediatric and non-pediatric departments – a claims data analysis.”

Reviewer 2 Report

The authors have carried out extensive analysis of pediatric chest injury. The main conclusion is that non-pediatric centers use more CT, which the authors believe is a potential area for reduction of radiation.

The authors need to make sure that pediatric and non-pediatric centers are not dealing with different groups of trauma patients of various severity before establishing the need for CT. The authors also need to maintain the interest the readers by explaining how the findings can improve the safety and management of pediatric chest injuries. 

Author Response

Point-by-point response to the comments of the reviewers

The authors would like to thank the reviewers for carefully reading our manuscript and providing valuable comments and suggestions to improve the paper. The following responses have been prepared to address all of the reviewers' comments in a point-by-point style.

Response to Reviewer #2:

We thank reviewer #2 for her/his comments. To acknowledge this valuable reviewer comment, we have amended:

1) “Table 8 (Supplement). Comparability of PD and non-PD with regard to the performance of CT” to the manuscript

2) the following sentence to the discussion in section Adult trauma centers use CT imaging more frequent than pediatric trauma centers: “To examine comparability of CT imaging between PD and non-PD, logistic regression for use of CT was performed (Table 8, Supplement). We estimated the same model as in Table 6 except that an interaction term of comorbidity and treatment department was added. The non-significance of the interaction term (P=0.2801) suggests that we can assume similar comorbidity patterns in PD and non-PD.”

3) the following sentence to the conclusion:Overall, our results emphasize the importance of centralization and implementation of evidence-based guidelines for specialized pediatric trauma care.”

Reviewer 3 Report

I want to congratulate the authors for their research. It was indeed interesting to read. I feel that a lot of patients sometimes miss out on the appropriate treatment because they do not get transferred to a specialised service and this article basically sums that up nicely. 

Author Response

Point-by-point response to the comments of the reviewers

The authors would like to thank the reviewers for carefully reading our manuscript and providing valuable comments and suggestions to improve the paper. The following responses have been prepared to address all of the reviewers' comments in a point-by-point style.

Response to Reviewer #3:

We thank reviewer #3 for her/his comments and are delighted that she/he congratulated us on our research.

Round 2

Reviewer 2 Report

How does the findings improve clinical practice and reduce morbidity and mortality remains to be elaborated 

Author Response

[Children] Manuscript ID: children-2184286

“Management of children and adolescents with chest trauma in pediatric and non-pediatric departments – a claims data analysis.” (Changed title of the paper after first revision; please see “Point-by-point response to the comments of the reviewers”)

Response to Reviever 2

The authors would like to thank Reviever 2 for carefully reading our manuscript and providing valuable comments and suggestions to improve the paper. The following response has been prepared to the comment: How does the findings improve clinical practice and reduce morbidity and mortality remains to be elaborated.”

We thank the reviewer for that critical question. The presented work represents a retrospective study describing the current management situation of children and adolescents with chest trauma in pediatric and non-pediatric departments in Germany based on an analysis of claims data. Main limitations are the missing information for clinical variables. However, due to the large sample size of 1 million children our results are very reliable. Our results emphasize the importance of centralization and implementation of evidence-based guidelines for specialized pediatric trauma care especially in adult trauma centers. This is an ongoing process, which is supported by our results. Thus, our findings may help to improve clinical practice and reduce morbidity and mortality.

To acknowledge this valuable comment, we have changed & amended the following the Conclusion section: “Our study describes the current management situation of children and adolescents with chest trauma in pediatric and non-pediatric departments in Germany based on an analysis of claims data. Main limitations are the missing information for clinical variables, e.g. the severity of the trauma (e.g. abbreviated injury score). However, due to the large sample size of 1 million children our results are very reliable. Thoracic CT imaging is significantly less frequently used in pediatric departments indicating a potential for reduction of CT imaging by implementation of evidence-based guidelines, especially in non-pediatric departments. Patients with a chest drain who are treated at a university/maximum care hospital show more injuries involving multiple body regions than patients with a chest drain managed at non-university hospitals without any difference in the length of hospital stay. This may indicate the effectiveness and quality of specialized pediatric trauma care in maximum care hospitals. However, since this subgroup accounts for less than 3% of patients, the clinical validity of this finding is questionable. Since 50% of children and adolescents with chest trauma are treated in adult departments our results emphasize the importance of centralization and implementation of evidence-based guidelines for specialized pediatric trauma care especially in adult trauma centers. To improve the quality and validity of future studies, claims data should be linked to clinical data.”

Round 3

Reviewer 2 Report

It’s helpful to explain how the findings may improve the management of pediatric chest injuries.

Author Response

[Children] Manuscript ID: children-2184286-revised_version_5

“Management of children and adolescents with chest trauma in pediatric and non-pediatric departments – a claims data analysis.”

 Response to Review Report (Reviewer 2)

 Reviewer 2: “It’s helpful to explain how the findings may improve the management of pediatric chest injuries.”

 @ Reviewer 2: Thank you, Reviewer 2, for providing valuable feedback. We have included the following sentences in the Conclusion section to acknowledge and address your comments appropriately.

…”However, this analysis alone does not have a direct impact on the current management situation.””In summary, our study indicates high-quality care for children and adolescents with chest trauma in Germany, as evidenced by the low mortality rate. To further optimization, we suggest implementing evidence-based guidelines, such as those to reduce the risk of ionizing radiation when using CT Thorax. However, linking claims data with clinical data is necessary to improve evaluation. Thus, our findings highlight the need for this connection in future studies.”

In the manuscrpit the changes are highlighted in turquoise.

We hope that we have now fulfilled your requests and high standards to your full satisfaction!

Kind Regards,

Peter Zimmermann MD PhD (corresponding author)

Department of Pediatric Surgery

University of Leipzig

Liebigstraße 20A

04103 Leipzig, Germany

Tel.: +49 341 97 26400

Fax: +49 341 97 26409

E-Mail: [email protected]